# Individual and Environmental Factors Associated with Participation in Physical Activity as Adolescents Transition to Secondary School: A Qualitative Inquiry

**DOI:** 10.3390/ijerph17207646

**Published:** 2020-10-20

**Authors:** Tomoko McGaughey, Janae Vlaar, Patti-Jean Naylor, Rhona M. Hanning, Lucy Le Mare, Louise C. Mâsse

**Affiliations:** 1BC Children’s Hospital Research Institute, Vancouver, BC V5Z 4H4, Canada; tomoko.mcgaughey@bcchr.ca (T.M.); janaev@student.ubc.ca (J.V.); 2School of Population & Public Health, The University of British Columbia, Vancouver, BC V6T 1Z3, Canada; 3School of Exercise Science Physical and Health Education, University of Victoria, Victoria, BC V8P 5C2, Canada; pjnaylor@uvic.ca; 4School of Public Health & Health Systems, University of Waterloo, Waterloo, ON N2L 3G1, Canada; rhanning@uwaterloo.ca; 5Faculty of Education, Simon Fraser University, Burnaby, BC V5A 1S6, Canada; lucy_lemare@sfu.ca

**Keywords:** physical activity, adolescent, school, environment, perceptions, transition period

## Abstract

The transition from elementary to secondary school is an emotionally and socially complex time when adverse behaviors appear, such as decreased levels of physical activity (PA). Behavioral and environmental factors that influence PA during this time are poorly understood. Therefore, we aimed to identify factors that influence PA as adolescents transition to secondary school. Qualitative interviews were conducted with a sample of 27 ethnically diverse child–parent dyads within the public-school system in British Columbia, Canada (50% boys, 68% mothers, 25% White). The interviews probed for environmental and behavioral factors in school, family, and social contexts that potentially initiated changes in PA, specifically related to the adolescents’ transitions. Interviews were recorded and transcribed verbatim. Thematic analyses identified factors at the individual, social, familial, and school levels that may trigger adolescents to change their participation in PA as they transition from elementary to secondary school. Twenty-two factors emerged from the qualitative analysis including school factors (8), household factors (3), social factors (4), and intrapersonal factors (7). These findings contribute to a better understanding of adolescents’ PA behaviors and highlight the influence of changing environments as they transition from elementary school to secondary school.

## 1. Introduction

In adolescents, participation in physical activity (PA) has numerous benefits, such as reduced risk of chronic diseases, improved metabolic profiles, healthier weight status, stronger bones, improved mental health and well-being, improved physical fitness (cardiovascular fitness, strength, and flexibility), better academic outcomes, and better brain health [1,2]. Despite the promotion of active lifestyles as a way to develop positive health outcomes and derive social benefits, decreasing levels of PA among adolescents are rapidly becoming a global phenomenon [3,4]. Approximately 90% of Canadian adolescents do not reach the benchmark of 60 min of Moderate to Vigorous Physical Activity (MVPA) per day [5]. Alarmingly, PA steadily declines between ages 9 and 15 years by as much as 60 to 75% [6,7,8].

Changes in adolescents’ behaviors are seen as the result of the relationship between the individual and their dynamic environment over time [9]. Importantly, major life changes such as the transition from elementary to secondary school mark a time when many environmental changes are taking place, which may affect adolescents’ PA [10,11]. Emerging evidence shows that individuals are more active during their elementary years than secondary school years [10,11]. In addition, recent evidence suggests that the transition to secondary school, especially when “it’s accompanied with a change in school, likely triggers the change and decline in PA that is seen in adolescence” [12,13,14,15,16,17,18]. Others have also noted that while overall PA declined after the transition, active transportation to school increased, although offset by a decrease in extracurricular PA [15,19,20,21]. A relatively small subset of adolescents remained active and increased their PA over time [11,15,16,17,18,19,20,21].

An agreed understanding of how the transition period from elementary to secondary school affects PA remains sparse, as the environments (school, family, peer, intrapersonal) adolescents experience vary drastically around the world [12,13,14,15,16,17,18,19,20,21,22,23,24,25,26]. In a previous qualitative study, girls’ participation in PA during the transition was related to the change in PA opportunities, decreased social support from friends and families to be active, increased pressure to be skilled in order to participate in PA, and competing priorities that took them away from PA [16]; however, changes like this are not as prevalent when regular MVPA is mandated in the school’s curriculum [18,19,20,21]. The transition to secondary school is often referred to as a pivotal event in an adolescent’s life as they experience numerous changes concurrently [18,19,20,21,22]. In the school setting, these changes include a new PA/Physical Education (PE) school environment [11,12,13,14,19]; a new range of school-based policies surrounding PA, PE, and the physical school setting that could affect the adolescents PA (i.e., availability of equipment) [15,17,18,19,20]; and differing expectations surrounding individual development and growth [21,22,23,24]. Changes in the household environment that have been identified to potentially modify PA during the transition are parental expectations for more responsibilities and independence [19,21,22,24,25,26,27]; and changes in the family’s role as a gate-keeper to adolescents PA [24,25,26,27,28,29,30,31,32]. Noted changes experienced in the adolescents’ dynamic social environment include the desire to develop new friendships, increased peer influence on the adolescent’s lifestyle [13,16,23,25,27,33,34,35], and the need for social support [25,26,27,28,29,34,35,36,37,38,39,40]. Finally, when reviewing our current understanding of adolescents’ experiences in changing PA during the transition from elementary school to secondary school, we see that during this time, adolescents increasingly desire to establish their personality and independence [41,42,43,44] including self-identifying if they enjoy PA [45], or if they feel it is important to their quality of life [42]. Although a number of studies have examined the factors that influence levels of PA after the transition, none have adopted a comprehensive socio-ecological approach when examining how multiple environments dynamically interact to affect PA behaviors as adolescents transition from elementary to secondary schools. Specifically, most studies have examined how a specific feature of the environment (such as friends or active travel) has changed during the transition and its relation to physical activity, but such an approach does not elucidate our understanding of the interrelations among diverse personal and environmental factors and PA during the transition to secondary schools. Understanding the interconnection among these factors will be essential to developing new interventions addressing changes in PA, as well as enabling better support for and management of current interventions so all factors that affect PA are addressed.

To address this research gap, we qualitatively examined how adolescents’ PA behaviors were shaped in relation to changes in the school, household, and social environments during the transition from elementary (Grade 7) to secondary (Grade 8) school. We sought to answer the following question: “What are the individual and environmental factors that influence adolescents’ involvement in PA as they transition to secondary schools.” A socio-ecological lens informed this research to reflect that there is a hierarchy of nested factors that influence adolescents’ PA. As such, a socio-ecological lens focused the understanding of the micro-level (individual factors such as motivation and interest), meso-level (social groups such as friends and family), and macro-level (environmental and societal such as the school environment or policy within the school) factors that influenced adolescents’ PA behaviors in secondary schools [10,46,47,48,49].

## 2. Materials and Methods

### 2.1. Participants

From March to May 2016, participants were recruited through a quota sampling approach from a large culturally and socio-economically diverse community in British Columbia, Canada. Ten schools were initially approached to ensure a proportional distribution of socio-economic status that would be representative of the community [50], which itself was selected because of its ethnic and socio-economic diversity. A socio-economically and ethnically diverse sample was selected to increase the generalizability of the findings to a wider group of families and ensure minorities were adequately represented. Once the schools consented to participate in the study, the study was advertised to Grade 7 students, through classroom presentations conducted by members of the research team. Twenty-eight 12–13-year-old adolescents and parents dyads participated in this study, each as independent sources of data to gain the perspective of change in PA from both the parent and adolescent. Demographics for these participants are presented in Table 1. Participants were eligible for inclusion if the adolescent was in Grade 7 (age 12–13), a parent was willing to participate in interviews as well, and both the parent and adolescent were fluent in English. Participants were not included if the adolescent reported a motor or mental impairment/disorder or health condition that limited participation in PA or severely restricted the foods the child could eat, or they had a psychiatric condition or substance abuse problem through the screener questions during the period of consent.

Interested participants contacted the research team and set-up a mutually agreeable time for an interview. Families were sampled in Grade 7 and sampling ensured an equal distribution of male and female adolescents (see socio-demographic characteristics of the sample in Table 1). These families were interviewed once in Grade 7 while the adolescent was in elementary school (March–May 2016), and the families were interviewed again after the adolescent transitioned to secondary school. Families who, after the transition, no longer met the eligibility criteria were excluded from the analyses. One family was excluded from the analysis as the child developed severe dietary restrictions during the transition.

### 2.2. Data Collection

Parent and child interviews were conducted separately but concurrently in the participant’s place of residence or in a community library at a date and time that was convenient. Interviews were conducted by trained research assistants, generally lasted from 30 to 75 min, and were digitally audio-recorded (Sony Linear PCM Recording Device—ICD SX712 and ICD700, Sony Electronics Inc, Tokyo, Japan). The interviews were semi-structured with probing when necessary. Questions asked during the interviews have been included in the Appendix A.

The prompting questions focused on the adolescent’s PA in Grade 8, how their PA had changed between Grade 7 and Grade 8, and probed to assess the individual and environmental factors (family, friends, and school factors) that the interviewees perceived as influencing current PA behaviors and how the transition had or had not changed PA behaviors. As these were semi-structured interviews, the questions were used as prompts, but interviewers allowed the participants to discuss topics they felt were important to them and deviate from the prompt if needed. This harmonized research study protocol was approved by the Research Ethics Board at the University of British Columbia (#H15–01876). All participants involved in the interviews provided informed consent to participate in the study.

### 2.3. Analysis

For this paper, only the Grade 8 interviews were analyzed and are presented. The 27 adolescent–parent dyads were each analyzed as a single unit so statements could be cross-referenced during the analysis. All interviews were transcribed verbatim and anonymized (names were removed from the transcripts) and given a unique ID that allowed the family transcripts to be linked for analysis. The open coding scheme was first developed using a sample of 10 families, and then reviewed for agreement by a research assistant who also coded the same sample of 10 families. Any discrepancies found were discussed with a third-party reviewer until agreement was reached. Once the coding scheme was found to be consistent and trustworthy, the remainder of the interviews were coded.

The qualitative thematic analysis was conducted using NVivo12 (QSR International, Doncaster, Australia) [51] as it allowed the dyadic transcripts to be linked and provided a systematic process for coding and a constant comparison analysis. The analysis included becoming familiar with the dataset and codes, searching for themes and factors from these codes, reviewing the generated data, defining and naming the identified themes and factors through a process of axial coding [52,53], and producing an end analysis based on the findings [52]. As noted previously, a socio-ecological lens guided the entire analysis process.

## 3. Results

Table 2 summarizes the various factors that were found to impact PA over the transition from elementary to secondary school and the different themes that emerged within those factors. Each of these is discussed below.

### 3.1. School Factors

Many factors within the school environment changed between elementary and secondary school and were identified as influencing adolescents’ participation in PA when they transitioned into Grade 8, including changes to:

#### 3.1.1. The Structure of Daily Schedule

A lack of designated allotted time and location to eat and no expectation to play after lunch influenced the adolescents’ decisions about whether they engaged in PA at lunchtime. This means that, in Grade 7, students who have designated times to eat subsequently play during their lunch time, but in Grade 8, there was no clear definition in how the student should be using their time. For example, Boy_1 says, “In Grade 7, there was a lunch monitor that made sure we ate our lunch, then after 15 min, we were allowed to go outside… now we can do whatever we want.” Another participant, Boy_9, mentioned that the lunch break was too short to incorporate PA “*we only have one 45-minute lunch break now whereas in elementary it would be, I think, seventy minutes total*.” In many cases, these changes meant that PA became driven by adolescents’ interests. Those who really wanted to be active figured out ways to incorporate PA into their new schedule, whereas those who were less inclined opted to pursue other interests as highlighted by Boy_10, “Last year my friends and I used to play tag during lunch. I liked tis because it was easy to play. Now [in Grade 8] some of my friends have started to play basketball during lunch, but I don’t like sports like that… I decided to try the photography club to fill my lunch time.”

#### 3.1.2. PA/Sports Opportunities

Sports opportunities also changed as some adolescents gained access to a weight room and/or had more sport choices within the secondary school setting. Secondary school provided opportunities for involvement in team sports that are generally competitive, such as football, rugby, and swimming. In some schools, opportunities were also provided for students to engage in less traditional team activities (e.g., ultimate Frisbee); however, the availability of these sports was dependent on the teachers’ willingness to support such a team. For some students, this increase in opportunities resulted in more PA, as noted by Boy_5 who said, “*I’d have to say a bit more active…. I guess it’s because you have more of a selection of sports kind of in the school*.”

#### 3.1.3. PA/Sports Accessibility

PA/Sports accessibility tended to change from ‘PA/sports for all’ in elementary school to a competitive model of making the team or needing enough skills to play a specific game in secondary school. While PA/sports opportunities generally increased in secondary school, these opportunities did not lead to an increase in PA for everyone. As mentioned by Girl_12, “*[The school] has a team, but they took like only five people… Too many girls tried out so I didn’t make it*.”

#### 3.1.4. Academic Expectations

For some students, the increase in academic expectations in secondary school over elementary school meant that they felt they did not have the time for PA. These participants mentioned the need to plan for their futures, and to develop themselves by focusing their efforts on academics and extracurricular activities like debate, model United Nations, and student counsel. In order to meet these demands, the participants felt they had to give PA less priority. As mentioned from Girl_2 “*I’m less active than last year…I tried to stay with everything, but it was too much… I was doing homework until like two o’clock…* (Girl_2).”

#### 3.1.5. How Physical Education (PE) is Assessed

Finally, the last major change experienced in PA within the school setting was how the Physical Education class was assessed. The three main factors that participants noted were:

1. Skills are assessed in secondary school.

After the transition from elementary school to secondary school, the participants noted that their PE classes changed from nonintensive, participation-based classes to graded, skills-based assessment. Participants felt that PE was less about play and having fun, and more about building and developing skills. This aspect motivated some students to be more engaged. For example, one boy reported, “*Yeah, in Grade 7 they didn’t expect much for physical activity but now that they [do] I actually like to do physical activity—they’re expecting like higher. I think it’s actually really good because I like expectations so I can get motivated* (Boy_2)”. However, it had the opposite impact for others. Boy_3 commented, “*I don’t like (PE) as much anymore our teacher makes us run a lot and its stressful being graded…we used to just play games.*”

2. There is a dedicated PE specialist in secondary school.

Another change from elementary school is that in secondary school, the PE teacher is a specialist rather than a generalist classroom teacher. Having a PE specialist teacher provided a healthy living role model for secondary students, and for some, their PE teacher was a source of support (e.g., taught strategies to be more active) and motivation to be physically active. Girl_3 commented, “*I think my teachers have the greatest influence because they are always encouraging saying oh you should go maybe walk around to like get exercise and stuff like that* (Girl_6).”

3. More time is needed for activity instructions in secondary school.

Along with the introduction of a PE specialist teacher, PE classes in secondary school were described as focused on skill development and improving sports knowledge. This generally involved a structured curriculum, instructional time, and test-based assessments. As a consequence, participants mentioned that they spent more time being idle in secondary school PE classes than they had in elementary school. This is reflected in the comment, “*Yeah we have PE class like every other day…when we are doing workouts or playing its fun but we spend a lot of time just standing there like that* (Girl_5).”

### 3.2. Familial Factors

For many parents we spoke to, their child’s transition from elementary to secondary school marked a time when they reassessed or changed their parenting practices related to PA, including:

#### 3.2.1. Unloading of PA Responsibility

For many families, the child’s transition from elementary to secondary school marked a time when parents reassessed or changed their parenting practices in relation to PA and off-loaded the responsibility for their child’s PA to someone else. Both child and parent participants mentioned the shift in responsibility to organize, facilitate, and initiate PA to:

1. The school

Shifting the responsibility of PA to the school including mention of the child joining more of the school sports, or engaging in school intramurals. Parents often noted the number of PA options that the school offered, which freed up the child’s time at home to focus on their academics. Reflecting a parental expectation of the school to fulfill her PA needs, Girl 3 noted, “*Both my parents encouraged me to join more sports (at school) because there are more sports to choose from* (Girl_3).”

2. Adolescents

Parents also noted that as a result of the transition, there was a need for the child to have more independence and ownership over their PA, either formal or informal. Parents seemed to feel this was an ideal time for the teens to take control over their own PA, to garner more independence in the teen, and to also foster some individuality by allowing this child to choose what sports they participate in. Expressing a shift in responsibility to the adolescent himself, the mother of Boy_7 reported, “*I don’t go to his practice anymore. I go do my own thing because I figure he’s old enough now. I’m hoping I can get him to a point where he doesn’t need me to drive him anymore so I can have more freedom*. (Mother of Boy_7).” Another parent mentioned, “*I want her to make her own choices, I know she did gymnastics and basketball before, but she seems to be preferring basketball more now… I’m not going to do gymnastics, it’s her choice.* (Father of Girl_1).”

#### 3.2.2. Family PA Time

Family PA time also changed because of changes in the school schedule, increased school expectations (e.g., more homework), and greater involvement in non-PA extracurricular activities (e.g., drama classes). For some families, this meant less PA time together, including evening walks or outings to the community center, as can be seen in the following quote: “*We used to go to the rec centre three times a week but now she is too busy with homework…we only go once in a while now* (Mother of Girl_7).”

#### 3.2.3. Independent Mobility

The shift to secondary school also meant changes in adolescents’ independent mobility (or active transportation without being with an adult). In some instances, this facilitated greater PA. For example, some adolescents gained the independence to walk to and from places without adults. In many instances, parents provided their adolescents with a mobile phone to facilitate this: “*They give me more freedom now because they trust me more…they let me walk to the rec centre on my own* (Boy_9).” However, others lost their independence as parents felt that the school was too far and they were not ready to provide this level of independence. As Girl_10 reported, “*No, I have to carpool to school now because it takes like forty-five minutes to walk there…Last year I was walking* (Girl_10).“

### 3.3. Social Factors

The transition from elementary to secondary school changed the social environment as many adolescents moved into larger schools within or outside their neighborhood. With this shift, social structures and norms changed as well.

#### 3.3.1. Friend Co-Participation

Among our adolescent participants, it was evident that physical activity was reinforced by co-participation with friends. Male adolescents noted that friend co-participation was a nice addition but was not a prerequisite to their participation in PA, as mentioned by Boy_7 “*A lot of the time I’ll bring a basketball from home and play before and after school, and during lunch. Sometimes my friends will join me, but I like just shooting hoops as well*.” However, females indicated that the engagement of their friends in PA was important to remaining active while also noting that they felt more comfortable socializing with their friends than being active with them. “*I never played basketball last year because my best friends didn’t like it but my new best friend and I play it all the time* (Girl_1).”

#### 3.3.2. Peer Validation of PA Skills

Peer validation of PA skills (or the feeling that peers evaluate whether you have physical activity skills) was less important in elementary school. Adolescents would interact with their peers as equals and were less competitive with each other during PE or recess. However, this collegial interaction disappeared in secondary school. Peer pressure encouraged adolescents to prove their PA skills and capabilities in a social setting. “*I don’t know but not to brag but I have always been fit, fast strong [my friends and I] used to just chase each other during lunch and play games just for fun but now the loser needs to buy the rest French fries. I usually win* (Boy_4).”

#### 3.3.3. Fitting in with Older Peers

The transition to secondary school introduced adolescents to older peers who were described as modeling a less active lifestyle. The desire to be socially accepted by older peers impacted their notions of wanting to be active and/or play. “*After school I usually sit around and hang out with my friends… they are older, like Grade 10, 11 and 12 and they don t like to run around and play* (Boy_13).” This quote illustrates how PA changed as social norms around play changed.

#### 3.3.4. Social Norms and Acceptability of Play

In elementary school, most adolescents noted that they were all relatively active, but the transition resulted in the formation of social groups with varied interests (e.g., artistic or nonactive group versus fit group), which dictated whether engagement in PA was more or less acceptable (i.e., play being no longer acceptable as this is for young children). “*We used to play during lunch but now we just hang out in the photography room or at the stage… I kind of want to do more but my friends just want to relax during lunch* (Boy_14).”

### 3.4. Intrapersonal Factors

Finally, there were a number of intrapersonal factors that influenced whether adolescents participated in PA after the transition to secondary school, including: 

#### 3.4.1. Skills and Aptitudes

How adolescents judged their own PA skills and aptitudes impacted their participation in PA more so in secondary than elementary school. In elementary school, PA was accessible to all, but after the transition, skills and aptitudes drove participation in PA and enjoyment in PE class. Some students enjoyed this challenge but others disengaged, identifying themselves as “no good’ or ‘not very good’ at the activity. This seemed to be related to the increased tendency for social comparison that comes with adolescence. As commented, “*I wasn’t worried about PE class in Grade 7 but now I just don’t want to make a fool of myself …everyone else can do the sport but I can’t* (Girl_8).”

#### 3.4.2. Motivation

Despite this tendency, there were others who were more motivated to be active in secondary school than they had been earlier. This appeared to be connected to their greater independence and was particularly apparent among males. “*I go riding with my friends to the library…I didn’t do it in Grade 7 but I am older now and more responsible…I didn’t want to before but now that I have the freedom to go I want to bike more* (Boy_11).”

#### 3.4.3. Athletic Identity

Adolescence is an important time for identity formation and it was apparent that some of the adolescents we interviewed had begun to value an athletic identity (i.e., having a strong desire to be physically active and be known for having specific physical activity skills). For some adolescents, it marked a time when their athletic identity was reinforced (i.e., being on the school sports team, excelling in PE, and playing sports in their free time at school) and helped them stay active. “*I joined the volleyball team and I want to join the basketball and rugby team…I like being fit* (Girl_9).”

#### 3.4.4. Desire to Form and/or Maintain Social Relationships

PA was also used as a way to form and/or maintain social relationships. Some adolescents noted that they used PA to make friends or they maintained their participation to keep their friendship with peers who had moved onto another school. “*I joined football to meet new people…I thought it would be a good way to make friends* (Boy_8).”

#### 3.4.5. Competing Interests and Preferences

The transition, and the exposure to other activities that it entailed, prompted adolescents to experience competing interests. Adolescents described that they explored new activities (PA or non-PA activities) as a way to develop their individuality like Girl_8 who mentioned, “*Like I said, everyone else can play sports like basketball or baseball, but I can’t… since my friends were busy playing, I started to get into other things to do like watch anime and draw and have made new friends that way….it makes me feel a bit better that I have friends I can relate to.*” This seemingly growing desire to be an individual plus new opportunities and resources allowed adolescents to explore their preferences, as demonstrated in this quote from Girl_1 “*I want to try everything and find what I like and what I’m good at. I know I like basketball so I’m going to keep practicing, but I also really like drawing, so I am looking forward to starting art in a few weeks. The teacher also runs an afterschool art club so I might join that..*.*I just want to be me and try everything*” When non-PA activities were of interest, adolescents had to divide their time between PA and those activities, whereas before, their free time, especially at school, would be dedicated to PA opportunities “*I decided to join extracurricular like the Reach Out and Student Council instead of joining the volleyball team. I wanted to try something new and I think it sounded like fun* (Girl_13).”

### 3.5. Factors Associated with PA Participation Varied by Context 

The factors associated with PA participation in secondary school varied depending on the context in which PA took place and whether adolescents: (1) Participated in organized PA/sports in or outside of school; (2) engaged in PA at school in their free time; and (3) engaged in PA in their free time at home or in the community. Table 2 summarizes the mechanisms through which the transition from elementary to secondary school may influence engagement in PA and highlights the factors that were relevant for a specific context. For example, PA parenting practices were mainly discussed in terms of how they influenced participation in organized PA (both in and out of school) and participation in PA in free time with friends and family, but were not mentioned in the context of PA participation in free time at school.

While engagement in PE is mandatory in secondary school, a number of factors discussed by the adolescents described whether they enjoyed PE and whether they were active during PE. Table 2 also highlights the factors that were mentioned to influence enjoyment of PE and participation in PA during PE. 

## 4. Discussion

This is the first study to provide an in-depth examination of how the transition from elementary to secondary schools is associated with adolescents’ participation in PA. Specifically, this study identified that: (1) The transition is a dynamic developmental period where intrapersonal factors and changes in the social environment (peers and family members) dynamically interacted to shape adolescents’ PA behaviors; (2) school PA practices in secondary schools appeared to meet the need of the emerging “athlete,” or those who were self-motivated to be physically active, which fostered the decline in PA for those who did not identify that way; and (3) the factors associated with PA participation after the transition are complex as adolescents’ PA participation was affected by factors at multiple levels (i.e., individual, interpersonal, and organizational), varied between adolescents as not all were affected similarly by the transition or the same factors, and moreover, they were context-specific (e.g., are different in the school and outside of the school context, for example). This study offers significant insight into the changes that occur during the transition and the impact these changes have on adolescents’ participation in PA. Ultimately, these findings could inform the development of interventions aimed at offsetting the decline in PA that occurs during the transition to secondary school.

We identified numerous intrapersonal factors and social factors within an adolescent’s environment that interacted in complex ways to influence volitional participation in PA. This finding aligns with a previous qualitative study conducted among girls where the researchers found a sense of self when active (such as being motivated to be active, enjoying PA, being confident in PA skills, and having PA skills judged by peers), and changes in social support from peers and family during the transition were related to their participation in PA [9,15]. In addition, it is well-established that the influence of peers increases during adolescence and influences adolescents’ PA [34,35,36,37,38,39]. Similarly, our study showed peers influenced PA participation after transition in a sample of boys and girls, but we also found athletic identity was in a state of flux during the transition. Essentially, adolescents’ sense of self as it relates to PA appears to be shaped by peers during the transition and began to settle during the first year of the transition. Once athletic identity, PA preferences, and involvement in competing interests become more settled, it does appear they set the stage for PA participation throughout adolescence, and for many, they initiate the decline in PA. This developmental process appeared to be quite dynamic during transition, influenced by both peers and intrapersonal factors in a reciprocal manner that is consistent with Bandura’s concept of reciprocal determinism [30]. These findings highlight the need to develop interventions that support the natural restructuring of social networks during the transition in a way that makes participation in PA the accepted social norm and desired identity.

Understanding the environmental factors that influence adolescents’ levels of PA during the transition from elementary to secondary school is complex as this study identified a number of intrapersonal and environmental factors that changed during this period. Given our understanding of the environmental factors that influence adolescents’ PA [16,17,18,19,20,21,22,23,24,25,26,27,28,29,30,31,32,33,34,35,36,37,38,39,40,41,42,54,55,56], it was not surprising that changes in the school, social, and household environments were highlighted as influencing adolescents’ PA during transition. Studies that examine factors associated with youth PA during the transition from elementary school to secondary school are relatively rare, and this paper addresses an important gap in the literature. This analysis highlighted the importance of considering how these various environments and intrapersonal factors interact to influence PA during the transition. Many of the environmental influences were themselves complex and impacted PA participation differently depending on adolescents’ intrapersonal factors and/or the influence of friends. This is not surprising, as adolescents’ enjoyment and motivation to be physically active are strong predictors of their PA participation [42,54], and many studies have also noted the influence of peers’ PA on adolescents’ levels of PA [34,41].

Parents play a central role in socializing their adolescents to PA opportunities, and this process starts when adolescents are young and continuously evolves as they age [23,24,43,55]. Similar to others, our study found that parents continued to play an important role in supporting adolescent PA [30,31,32,33]. However, from the qualitative analyses, we identified three key changes that were either frequently mentioned by the participants or were described to have the largest impact on the adolescents’ PA. These factors were the unloading of PA responsibility, reduced family PA time, and practices related to independent mobility. From previous studies, we know that less family PA time or restricted independent mobility affects PA and/or active transportation [27,28,29,30,31,32,33]. Our qualitative analysis identified the transition as a time when some parents opted to unload responsibility for PA to either the school or their adolescent. This may occur because, over time, PA has emerged as a highly programmed activity and routine that consumes parents’ schedules when adolescents are younger [30,55]. While many parents expected the school environment to meet their adolescent’s need for PA, the interviews uncovered that the school only partly met this need as it became mainly available for those who were skilled, could become part of the school team, had the time to take in these activities, or were motivated to take part in PA.

The school environment included several changes related to PA including more resources (equipment and space), increased sports opportunities, and having staff who were dedicated to teach/coach PA. From the school perspective, there was an increased investment in PA and PE as it became integrated within the educational mandate as a required graded subject. Our analyses indicated that there were several aspects of PA and PE provision in secondary schools that negatively affected adolescents’ participation and enjoyment in PA. This finding is consistent with other studies that found grading PE negatively impacted PA enjoyment [57,58]. To engage adolescents in PA at school, this study identified the need to: (1) Provide alternative sporting or PA opportunities for those who did not qualify for team sport; others have noted that adolescents in schools that provide more intramural opportunities accumulated more total and vigorous PA per week [59]; (2) reimagine the structure of PE class to retain PA engagement among those who may feel that they lacked skills or self-esteem to be PA among their peers; and (3) rethink how the PE period is used to ensure that adolescents are physically active during PE as the traditional way of teaching PE (e.g., passive instructional teaching and practicing one at a time), which has been found to minimally contribute to PA in PE among students (~16 min per PE session) [59,60,61]. Given that PA remains an important aspect of the educational mandate and as PA is associated with academic achievements [60], it is important to address the issues that negatively influence adolescents’ participation in PA as they transition into secondary schools.

In this study, numerous factors were found to affect adolescents’ participation in PA. Importantly, attempts to counteract the decline in adolescent PA need to account for the complexity of factors that are at play during the transition across individual, social, and school levels. While many of the adolescents interviewed identified barriers to sustaining PA engagement after the transition, a small proportion of adolescents were quite successful at remaining engaged and, with respect to PA, flourished in their new environments. This observation is consistent with a longitudinal study where a significant proportion of adolescents were found to remain physically active after the transition and even continued to increase their participation over time [15]. Our observations provide some extended insight into factors leading to this high engagement; among others, these adolescents seem to start secondary school with an established athletic identity and strong PA skills. It may be that this aspect drove their motivations, preferences, and participation in sports and PA, as athletic identity has been linked with adolescent PA [51]. While it is helpful to understand how some adolescents remain active and continue to be active as they grow, it is likely that this sub-group of adolescents are influenced by inner factors and unlikely influenced by other factors related to the collective environment.

As with all research, this study is not without its limitations. First, recall of PA behaviors in elementary school may have been biased and influenced by social desirability. This bias was somewhat mitigated by analyzing the adolescents’ and parents’ transcripts as a unit that served to cross-reference statements and provide a more complete understanding of actual behaviors. Second, the participants did not have the opportunity to validate the factors that emerged from the analysis, as it was not completed the same year the data were collected, and parents and adolescents’ perspectives may have evolved or changed. Third, the families who volunteered to participate were likely not representative of the general population, which could result in bias and misrepresentation in the general student population. In addition, it is important to highlight that the social and cultural context in which the data were collected limits the generalizability of the factors identified. It is likely that the findings are most relevant in countries that have similar cultural context and school structure. Fourth, the interviews excluded discussion of more macro-level factors and societal factors (e.g., mass media, marketing, and advertising) that can affect PA behaviors [9,10,11]. Fifth, as researchers carry out the qualitative analysis, it is important to consider that personal, cultural, and knowledge of the field of PA may have biased the interpretations. These factors are also important and should be added in future studies. While this study has a number of limitations, its major strengths include: (1) Having a large diverse sample of adolescents and their parents (54 interviews); (2) conducting separate interviews with adolescents and parents, which allowed for a more well-rounded representation of the changes experienced during the transition; and (3) having representation from adolescents across a wide range of SES and ethnicities to ensure the changes reported by the adolescents were more universal.

## 5. Conclusions

This study highlights the factors that interact and contribute to the changes in adolescent PA experienced in secondary school. Addressing drops in PA during adolescence that are associated with their transition to secondary school is critically important as the immediate physical, social, and mental health benefits and importance of these habits to adult health are well-established. We show the complex relationship of factors across multiple levels that interact with context. It is important to address parental, individual, and school-level factors. Through this emotionally and socially complex period, it is important that adolescents maintain or increase their levels of PA as they transition to secondary school to ensure healthy behaviors leading into adulthood. Future research should focus on developing solutions to address the comprehensive list of factors we identified to influence adolescents’ PA and ultimately decrease the decline in adolescence. By acknowledging the factors noted in this paper, sustainability of PA into late adolescent and adulthood can hopefully be achieved. This qualitative analysis provides the foundation for further quantitative research examining the changes in PA during the transition from elementary school to secondary school, in order to create a future that will support adolescents during this dynamic period.

## Figures and Tables

**Table 1 ijerph-17-07646-t001:** Participant characteristics (*n* = 27) ^1,2^.

Variable	Characteristic	% or Mean (Standard Deviation)
Adolescents Gender	Female	48.0
Adolescents Age (years)		13.2 (0.5)
Parent Gender	Female	67.0
Parent Age (years)		43.9 (9.0)
Parent Ethnicity	White	26.0
	East/South East Asian	26.0
	South Asian	33.0
	Other (Jewish, West African, Middle Eastern)	15.0
Parent Education	High School or less	22.0
	Trade/Vocational/College Diploma	26.0
	Bachelor Degree	22.0
	Above Bachelor Degree	30.0
Parent Marital Status	Married/Common Law	78.0
Household Income	$40,000 or less	19.0
	$40,001–$80,000	41.0
	$80,001–$120,000	21.0
	$120,001 and above	19.0

^1^ Participant characteristics were collected when the adolescents were in Grade 7 but the qualitative data in this paper were collected when the adolescents were in Grade 8; ^2^ the information of one participant was not included, as they did not meet the study requirements during Grade 8.

**Table 2 ijerph-17-07646-t002:** Mechanisms through which the transition from elementary to secondary school influences physical activity (PA) in various settings (*n* = 27 families).

Reported Change	Factor		PE Context		Participation in			
			Enjoy PE	Active in PE	Organized PA/Sports in School	Organized PA/Sports Outside of School	Participation in PA/Sport in Free Time at School	Participation in PA/Sport In Free Time with Family/Friends
**SCHOOL FACTORS**								
~	Structure of daily schedule							
▲	PA/Sport opportunities							
**New**	PA/Sport accessibility							
▲	Academic expectations							
**New**	Physical Education (PE) becomes graded	Skills are assessed						
		Dedicated PE specialist						
		Need time for instructions						
**FAMILIAL FACTORS**								
=/~	Unloading of PA responsibility	School						
		Adolescent						
▼	Family PA time							
=/~	Independent mobility							
**SOCIAL FACTORS**								
~	Friend co-participation							
▲	Peer validation of PA skills							
~	Fitting in with older peers							
~	Social norms and acceptability of play							
**INTRAPERSONAL FACTORS**								
~	Skills and aptitudes							
~	Motivation							
**New**	Athletic identity							
**New**	Form and/or maintain social relationships							
**New**	Competing interests and preferences							

**Legend.** ▲ factor increased with the transition; ▼ factor decreased with the transition; = factor remained the same with the transition; ~ factor changed with the transition; =/~ factor remained the same for some and changed for others; NEW factor is a new concept that is triggered by the transition. When multiple symbols are used, it denotes that the families experienced the changes differently. Factors that influence a given setting. Grey shading indicates the factor contributes to changes experienced in the indicated modified outcome.

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
