# Peer review of "Individual and Environmental Factors Associated with Participation in Physical Activity as Adolescents Transition to Secondary School: A Qualitative Inquiry"

_ijerph, 2020, doi:10.3390/ijerph17207646_

Round 1

Reviewer 1 Report

This manuscript is exceptionally well-written.  Only minor grammatical errors are noted:  P. 5/15, line 211”this children’s” should read, “those children”.  P. 11/15, line 374, “…to be physical active” should read, “…to be physically active”.   P. 11/15, line 376, delete the redundant qualifier, “especially in adolescence”.  P. 12/15, line 386, “parents; schedules” should read “parent’s schedules”.

Publication of this manuscript will hopefully inspire similar study in other parts of the world.  Perhaps not all twenty-two factors will be of similar relevance in other cultures.  Approaches to mitigate and reverse the three key factors that negatively affect physical activity in transition from elementary to secondary school (Page 12/15, lines 380-391), will possibly require cultural contextual nuances especially for girls.  The solutions (Page 12/15, lines 394-409) might be relevant almost universally).  The authors offer thoughtful consideration of limitations to the generalizability and potential bias, but not from a global perspective. 

Author Response

Comment 1: This manuscript is exceptionally well-written.  Only minor grammatical errors are noted:  P. 5/15, line 211”this children’s” should read, “those children”.  P. 11/15, line 374, “…to be physical active” should read, “…to be physically active”.   P. 11/15, line 376, delete the redundant qualifier, “especially in adolescence”.  P. 12/15, line 386, “parents; schedules” should read “parent’s schedules”.

REPLY: Thank you for the positive feedback.  All these edits were made in the manuscript.

Comment 2: Publication of this manuscript will hopefully inspire similar study in other parts of the world.  Perhaps not all twenty-two factors will be of similar relevance in other cultures.  Approaches to mitigate and reverse the three key factors that negatively affect physical activity in transition from elementary to secondary school (Page 12/15, lines 380-391), will possibly require cultural contextual nuances especially for girls.  The solutions (Page 12/15, lines 394-409) might be relevant almost universally).  The authors offer thoughtful consideration of limitations to the generalizability and potential bias, but not from a global perspective.

REPLY: We have expanded our limitation section to address this and added the following statement on page 14 and line 478-482 “In addition, it is important to highlight that the social and cultural context in which the data was collected limits the generalizability of the factors identified. It is likely that the findings are most relevant in countries that have similar cultural context and school structure.”

Reviewer 2 Report

I'd appreciate the opportunity to review this work. I've attached the original pdf file with highlights and comments in the work to be answered when applicable. Most part are suggestions, that would enrich the work to a better understanding and appreciation of the study. 

Best regards, 

Author Response

REVIEWER 2

Comment1: on page 1 / 15, line 24: (Interviews) Will the template for the interview be available as a sup. material? REPLY: We have added the interview scripts for parents and adolescents as supplementary documentation.

Comment 2: on page 2 / 15, line 82: (one of their parents) As a subject? REPLY: Yes, the parents were also used as subjects. To clarify this, the statement on page 3 / 17 line 97-98 was changed to “one of their parents participated in this study, each as independent sources of data to gain the perspective of change from both the parent and adolescent”

Comment 3: on page 2 / 15, line 86: (were excluded) I'd suggest "Not included" instead. REPLY: This change was made.

Comment 4: on page 2 / 15, line 86: (reported) I'd suggest to add "....in the screen visit". REPLY: This sentence was rewritten.

Comment 5:  on page 2 / 15, line 86: (disability) I'd suggest better specify: e.g. "motor or mental impairment/disorder" that limited... REPLY: This change was made

Comment 6: on page 2 / 15, line 87: (influenced dietary intake) What does that means? Higher BMI in dietary intake higher than XXX kcal/meal? REPLY: This statement was clarified.

Comment 7:  on page 2 / 15, line 87: (psychiatric)  I'd suggest: "...a psychiatric condition or..." instead of "..had a psychiatric or" REPLY: This change was made.

Comment 8: on page 2 / 15, line 88: (required learning assistance) Which points to a mental disorder;said before. REPLY: The statement was removed.

Comment 9: on page 2 / 15, line 88: (during) during the period of the consent. REPLY: We modified the sentence as suggested.

Comment 10: on page 2 / 15, line 91: (then again, a year later) So here we have a case they could be excluded during the period of the study (between pre- post-) and I'd suggest to add exclusion factors (and there was 1)  e.g. physical accidents limiting PA or other similar conditions. REPLY:  We clarified the grade 8 exclusion factors and added the following text on page 3 and lines 108-117 “These families were interviewed once in Grade 7, while the adolescent was in elementary school (March - May 2016) and the families were interviewed again when the adolescent transitioned to secondary school. Any families who were discovered to no longer meet the eligibility criteria were excluded from the analyses. One family was excluded from the analysis as the child developed severe dietary restrictions during the transition.”

Comment 11: on page 3 / 15, line 94-95: (only the Grade 8 interviews were analyzed and presented) Table 1 says: "1-Participant characteristics are from the Grade 7...".  I'm confusing if the statement "for the purpose of this report" points to the Table 1, or the entire data set. I'd suggest rephrase that. Yet, the analysis information could be moved to "Analysis" section. REPLY: As suggested the statement was moved to the analysis section and rephrased (see page 3. lines 108-117). In addition, we further explained the footnote in Table to state that the participants’ characteristics were collected when the adolescents were in Grade 7 but the qualitative data reported in this paper was collected in Grade 8 (see Table 1).

Comment 12: on page 3 / 15, Table 1: (Other) I'd suggest to state somewhere in the methods section the "Other" classification. I'd like to know what is considered "other" ethnicity compared to White and Asian. However, in case there is only two or three (I don't think would be more than that) would be acceptable adding two line in the table, I'm pretty sure it would be still good. REPLY: This information was added to Table 1.

Comment 13: on page 3 / 15, Table 1: (Parent Education) If available to the authors, the physical activity level/ h/week PA from the parents could be stated. Reply: This information was not collected so this was not added to Table 1.

Comment 14: on page 3 / 15, Line 105: (where) would it be "when" ? REPLY: This change was made.

Comment 15: on page 3 / 15, Line 113: (anonymized) Have the individuals received an random ID and then that were paired analyzed? Reply: We explained what we meant by anonymized and clarified how the transcripts from parents to adolescents from the same family were linked.  “...anonymized (names were removed from the transcripts) and given a unique ID that allowed the family transcripts to be linked.”

Comment 16: on page 3 / 15, Line 113-116: (The open coding scheme was first developed by author TM using a sample of 10 families, and then reviewed for agreement by a research assistant (RA) who also coded the same sample of 10 families) I'd suggest just say e.g.: subjects/families were randomized into numeric ID's .Authors' contribution can be stated in a separated section after conclusion section. Any discrepancies found were then presented to a third-party reviewer (author LCM) until everyone agreed. REPLY: We eliminated this information as the contribution section provides this information.

In regards to the authorship comments, I have included the description

Comment 17: on page 3 / 15, Line 118: (NVivo12) Which type of statistical analysis? Reply: we clarified that this is a qualitative thematic analysis.

Comment 18: on page 5 / 15, Line 194-197: (For both the child and parent participants, they mentioned this including shifting the responsibility to organize, facilitate, and initiate PA to) Same sentence. REPLY: The second sentence was deleted.

Comment 19: on page 6 / 15, Line 243-245: (A lot of the time I’ll bring a basketball from home and play before and after school, and during lunch. Sometimes my friends will join me, but I like just shooting hoops as well) italic, just to keep consistence: REPLY: This change was made.

Comment 20: on page 9 / 15, Table 2: (Academic expectations) Just trying to figure it out the way the information is present. So in this case (highlighted) increase in Academic expectations increase the PA in all PA conditions at right? Or the academic expectations, that could be either higher or lower, influenced increasing PA on that given conditions? In case of equal or unchanged status, so it means the primary condition (lets call primary factor) as in "e.g. independent mobility" primarily influences only free time PA with friends (makes sense), but doesn't increase or decrease the level of PA. I guess it would be clearer using a score for the questionnaire's answers. Such as sum points, ranking or "yes" "no", Chi-square would work well on this case. REPLY: The information from this table was obtained from the qualitative transcripts and not from questionnaire data.  We clarified that the legend denotes whether families talked about whether the factor increased, decreased, remained the same or changed.  When multiple symbols are shown in denotes that these factors did not change in the same way for all the families interviewed.  We clarified this in the legend section of Table 2.

Comment 21: on page 9 / 15, Table 2: (Unloading of PA responsibility) I think percents of N would help. REPLY: As we clarified in comment 20, the information presented into Table 2 is from qualitative interviews and as such it is appropriate to report percentages.  We modified Table 2 to include the N for the analyses.

Comment 22: on page 12 / 15, lines 380 - 382: (we identified three key changes that happened during the transition that negatively impacted PA, namely unloading of PA responsibly, reduced family PA time, and practices related to independent mobility) I'd suggest rephrasing that:

There was no statistics or calculations behind the classification of higher to lower impact. *At least not present. Even though having a higher incidence in the study case amongst the participants, I strongly recommend disclose that and better compute the ratios in the given sample. Reply: We rephrased this section to highlight that this statement is made in the context of a qualitative study.

Comment 23: on page 12 / 15, lines 380 - 382: (some parents opted to unload PA responsibilities) How many percentage? REPLY: We clarified that the statement is in the context of a qualitative analyses and we did not add percentage as percentages are not considered relevant or stable with qualitative studies.

Comment 24: on page 12 / 15, lines 390: (wide variety of PA opportunities) Depending on the country on question, I believe there will be a wider variety of sports and activities and also the chance to do it. I'm not sure the colleges in question in Canada, but the statement gives the idea that the lack of PA variety and opportunity to play, influences the given sample in any degree for choosing or not playing any sport. For instance, as a certain % of the sample stated performing less PA due to the lack of favorite PA. In matter of fact it can happen, but seems to me that the load of homeworks and free time itself was a greater motivator. However, again, numbers would better describe the sample, e.g. "95% of the subjects has complained as the main factor not doing and PA is because they don't have freetime", them we know what is the key-effect. REPLY: We expanded our explanation to indicate that time to take part in these activities and being motivated to be physically active could also explain the decrease in PA. As indicated above we opted to not report percentage given the qualitative nature of this study.

Comment 25: on page 12 / 15, lines 390: (lacked skills or self-esteem) Was the sample classified by skill set or self-esteem level in any ranking manner? REPY: The self-esteem was based on the child’s description of their skill set or confidence surrounding doing PA.  The statement on page 13 / 17 lines 475 – 476 to : “2) reimagine the structure of PE class to retain PA engagement among those who may feel that they lacked skills or self-esteem to be PA among their peers;”

Comment 26: on page 12 / 15, lines 413-414: (a small proportion of adolescents were quite successful at remaining engaged and with respect to PA flourish in their new environments) and assuming they have the same curriculum, PA access opportunities, variety and climate, etc, it points out to an inner factor not strongly related to the collective environment. REPLY: We modified our concluding statement to say “….this sub-group of adolescents are influenced by inner factors and unlikely influenced by other factors related to the collective environment.”

Comment 27: on page 13 / 15, lines 434-435: (conducting separate interviews with adolescents and parents) a match rate of accuracy would be a good idea. REPLY: We clarified in the participant section and the analysis section that the sample includes 27 parent-adolescent pairs and the data was analyzed as a unit of analysis.  

Comment 28: on page 13 / 15, lines 439: (hierarchal factors) Hierarchal brings the notion of levels of influence, thinking of the factors. And the study doesn't provide (at this moment) enough information to classify levels of influence. REPLY: We eliminated this term.

Comment 29: on page 13 / 15, lines 440: (they transition) implies pre post analysis. REPLY: We modified the sentence and we are now refereeing to PA experienced in secondary school.

Reviewer 3 Report

Dear Authors, 

This work is commendable. You are right, there is a scarcity of this depth of information. However, the study would benefit from more than just the research and results. I would recommend:

1) There was a lack of a strong review of the literature. Please add

2) There was not much grounding of the work. Consider adding more about this research, gaps it sought to fill, clear hypothesis, theoretical framing and grounding, more about the population and choosing this sample etc.

3) A listing of what was examined under each category would have been very helpful. Yes, school, social, and other environments, but what do these mean? What terms are used to define them? How did you determine those factors, under each, were sufficient to give a complete picture of that category?

4) What were the limitations? What biases did the researchers bring to the work?

5) A definition of the final terms or factors from the analysis would have also been effective for determination on relevance generalization.

6) More about what qualitative and the software worked best to capture this data versus prior methods and limitations of other methods and study would have ensured a strong manuscript too.  

Publishing this work is important. Addressing these concerns will yield such result.

Author Response

Comment 1: This work is commendable. You are right, there is a scarcity of this depth of information. However, the study would benefit from more than just the research and results. REPLY: Thank you for your feedback.

Comment 2: There was a lack of a strong review of the literature. Please add.

REPLY: We expand the introduction section to explain the gap this study filled. We added the research question that this qualitative study sought to answer. We added the following to page 2 lines 55-78:

“An agreed understanding of how the transition period from elementary to secondary school affects PA remains sparse, as the environments (school, family, peer, intrapersonal) adolescents experience vary drastically around the world [12-26].  In a previous qualitative study, girls’ participation in PA during the transition was related to change in PA opportunities, decreased social support from friends and families to be active, increased pressure to be skilled in order to participate in PA and competing priorities that took them away from PA [16], however changes like this are not as prevalent when regular MVPA is mandated in the school’s curriculum [18-21]. The transition to secondary school is often referred to as a pivotal event in an adolescents’ life since they experience numerous changes concurrently [18-22]. In the school setting, these changes include a new PA/Physical Education (PE) school environment [11-14, 19]; a new range of school-based policies surrounding PA, PE, and the physical school setting that could affect the adolescents PA (i.e. availability of equipment) [15-17, 20]; and differing expectations surrounding individual development and growth [21-24]. changes in the household environment that have been identified to potentially modify PA during the transition are expectations as parent may provide their adolescents with more responsibilities and independence [19, 21, 22, 24-27]; changes in the family’s role as a gate-keeper to adolescents PA [24-32]. Noted changes experienced in the adolescents  dynamic social environment includes the desire to  develop new friendships and increased peer influence in the adolescent’s lifestyle [13, 16, 23, 25, 27, 33-35], need for social support; [25-29, 34-40]. Finally, when reviewing our currently understanding of adolescent’s experiences in changing PA during the transition from elementary school to secondary school, we see that during this time, adolescents see a desire to establish their personality and independence [41-44] including self-identifying if they enjoy PA [45], or if they feel it is important to their quality of life [42].”

Comment 3: There was not much grounding of the work. Consider adding more about this research, gaps it sought to fill, clear hypothesis, theoretical framing and grounding, more about the population and choosing this sample etc. REPLY: We expanded the introduction to explain the gap this study filled.  We added the research question that this qualitative study sought to answer.  Given that this is a qualitative study, it is not appropriate to add study hypotheses but instead we provided our research question. We further explained our theoretical lenses for the qualitative analysis. We added the rationale for selecting our study site in the participant section.

Comment 4: A listing of what was examined under each category would have been very helpful. Yes, school, social, and other environments, but what do these mean? What terms are used to define them? How did you determine those factors, under each, were sufficient to give a complete picture of that category? REPLY: We opted to describe the lens that we used in the analysis but given the nature of the study being qualitative we cannot a-priori state an hypothesis and we used a thematic analysis to uncover the factors within each category.  We now provide some examples in the description of our theoretical lenses but refrain to define those as this is the purpose of the qualitative study which is to define those categories.

Comment 5: What were the limitations? What biases did the researchers bring to the work? REPLY: As researchers carry out the qualitative research and are integral into processing and synthesizing the data, it is impossible identify the bias that each researcher brings. To minimize biases of individual researchers, we used a transparent and critically reflective processes. In addition, the information that was extracted and synthesized from the transcripts were triangulated.  While we cannot guarantee that researcher biases were fully eliminated with our process we expanded the limitation section to address this concern.

Comment 6: A definition of the final terms or factors from the analysis would have also been effective for determination on relevance generalization. REPLY: We added definitions to factors that might not be intuitive (e.g., athletic identity) and did not add definitions to factors that were self-explanatory.

Comment 7:  More about what qualitative and the software worked best to capture this data versus prior methods and limitations of other methods and study would have ensured a strong manuscript too. REPLY: We added the rationale for using the NVivo software for the analysis and added that we used a thematic analysis and a constant comparison approach to summarize the data.

Comment 8: Publishing this work is important. Addressing these concerns will yield such result. REPLY: Thank you for the kind feedback.

Round 2

Reviewer 2 Report

I appreciate the opportunity in reviewing the presented work. I could notice the authors have work extensively on modifications improving the manuscript, modifications that have indeed enhanced the scientific soundness and quality of presentation. My best wishes to the authors and congratulations for the work. 

Best regards, 

AF